# Unusual nuclear exchange within a germanium-containing aromatic ring that results in germanium atom transfer

Ryohei Nishino [1], Norihiro Tokitoh [1,2] ✉, Ryuto Sasayama[1], Rory Waterman [3] & Yoshiyuki Mizuhata [1,2] ✉

The delivery of single atoms is highly desirable for the straightforward synthesis of complex molecules, however this approach is limited by a lack of suitable atomic transfer reagents. Here, we report a germanium atom transfer reaction employing a germanium analogue of the phenyl anion. The reaction yields a germanium-substituted benzene, along with a germanium atom which can be transferred to other chemical species. The transfer of atomic germanium is demonstrated by the formation of well-defined germanium doped molecules. Furthermore, computational studies reveal that the reaction mechanism proceeds via the first example of an aromatic-to-aromatic nuclear germanium replacement reaction on the germabenzene ring. This unusual reaction pathway was further probed by the reaction of our aromatic germanium anion with a molecular silicon species, which selectively yielded the corresponding silicon-substituted benzene derivative.

In organic synthesis, the development of synthetic strategies relies on dividing the target molecule into simpler fragments (i.e., retrosynthetic analysis). The difficulty of a synthesis is often correlated to the number of the steps, and the accessibility of the compounds depends on the availability of requisite building blocks. Single atoms are the simplest building block of all molecules, however for most elements access to individual atoms for synthesis is impractical. This drives increased complexity, lower overall yield and poor atom economy in chemical synthesis. For compounds containing group 14 elements, the lack of available atomic synthons hampers synthesis. In recent years, progress has been made in understanding and utilizing molecular sources of elemental forms of these elements. A significant advance was the isolation of formally E(0) (E = Si, Ge, Sn, and Pb) compounds, termed metallylones, in which the central E(0) atom is coordinated by Lewis bases[1-7]. There are also reports of the related dinuclear compounds such as disilicon and digermanium[6,8-11]. While the nature of the bonding in these compounds is still being investigated[12-14], some such species exhibit reactivity which demonstrates their use as single Si, Ge, and Pb atom sources (Fig. 1a)[15-19]. For

example, Wesemann and co-workers reported the synthesis of phosphine-stabilized digermavinylidene[15] and germasilavinyidene[16]. The natural resonance theory (NRT) analysis of these compounds revealed minor resonance contributions of the ylidone structure which possesses dative bonds L → E ← L (L = ligand), highlighting their potential utility as monoatomic-Si or -Ge synthons. Indeed, the reaction of these compounds with diimines or azides resulted in the formation of N-heterocyclic silylenes (NHSi) or $N_4Ge_4$ cubane-type clusters, respectively. Novel molecular architectures and improved synthetic efficiency are expected to be realized from atomic synthons of the heavier group 14 elements.

We have been studying aromatic compounds containing one or more skeletal heavy group 14 element (Si, Ge, and Sn) in so-called heavy benzenoids. They are extremely reactive and easily undergo auto-oligomerization. By employing bulky protecting groups such as Tbt (2,4,6-tris[bis(trimethylsilyl)methyl]phenyl), we have succeeded in synthesizing and isolating these as thermally stable compounds[20]. Recently, we demonstrated that the treatment of Tbt-substituted germa- or stannabenzene with $KC_8$ or alkali metal naphthalenides

[1]Institute for Chemical Research, Kyoto University, Gokasho, Uji, Kyoto 611-0011, Japan. [2]Integrated Research Consortium on Chemical Sciences, Gokasho, Uji, Kyoto 611-0011, Japan. [3]Department of Chemistry, University of Vermont, Burlington, VT 05405-0125, USA. ✉e-mail: tokitoh@boc.kuicr.kyoto-u.ac.jp; mizu@boc.kuicr.kyoto-u.ac.jp

**Fig. 1 | Atomic synthons of group 14 elements. a** Low-valent compounds exhibiting single-atom transfer reactions. **b** Reaction of potassium germabenzenide **1** with dibromodigermene **2-Ge** which affords germabenzene **3-Ge** and molecular germanium clusters via Ge atom transfer.

resulted in the elimination of Tbt group to give heavy analogs of phenyl anion, i.e., germa[21,22] and stannabenzenyl anions[23]. It is notable that these heavy phenyl anion analogs are thermally stable despite a lack of steric protection. As demonstrated by their reactions with Cp*RuCl[24] and chlorosilanes[25], these heavy phenyl anion analogs have found utility as nucleophiles leading to new syntheses. During the investigation of the reaction of germabenzenyl anion **1** with 1,2-dibromodimetallenes, Tbb(Br)Ge=Ge(Br)Tbb (**2-Ge**) and Tbb(Br)Si=Si(Br)Tbb (**2-Si**, Tbb = 4-*tert*-butyl-2,6-bis[bis(trimethylsilyl)methyl] phenyl), we uncovered an unusual germanium atom transfer reaction (Fig. 1b). The observed reactivity is ascribed to an unusual Ge atom replacement on the aromatic germabenzenyl ring to the dimetallene-derived Ge or Si atom, giving the corresponding germa- (**3-Ge**) or silabenzene (**3-Si**), which provides insight into the atom transfer process. Such heteroatom substitution chemistry is reminiscent of pyrylium salts ($C_5R_5O^+$), which are isoelectronic to phenyl anions and undergo a variety of oxygen atom replacement reactions, allowing for the synthesis of various aromatic heterocycles[26].

## Results and discussion

### Reactions of potassium germabenzenide 1 with 1,2-dibromodigermene 2-Ge

In continuation of our studies examining the use of the potassium germabenzenide **1** as a nucleophile, **1** was combined with the 1,2-dibromodigermene **2-Ge** bearing Tbb groups[27] in THF solution (Fig. 2a). This initial reaction yielded several compounds, but upon solvent exchange to benzene-$d_6$ followed by heating and exposure to ambient light, a mixture of germabenzene **3-Ge** and the Ge/C cluster **5-2** was ultimately obtained. The observation of the germabenzene **3-Ge** bearing a Tbb group was quite unexpected as it suggests the germanium atom in the aromatic ring is derived from **2-Ge** rather than **1**. Crystallization of the reaction mixture from hexane afforded **5-2** as

blue-green crystals in 19% yield which were fully characterized. The reaction initially afforded three compounds: germabenzene **3-Ge**, digermabenzenylgermyl anion **4**, and **5-1**. Subsequent heating of the mixture resulted in the conversion of **4** to **3-Ge** and **5-1**. The conversion of **5-1** to **5-2** was observed upon exposure to ambient light (Supplementary Figs. 26–28). Each **3-Ge**, **4**, and **5-1** could be prepared independently to unambiguously confirm their identity. The isolated compound **5-1** is completely unchanged under thermal conditions in the dark (75 °C in $C_6D_6$) and isomerizes to **5-2** only when exposed to ambient light.

Although the formation mechanism of **5-1** is not clear, both Ge/C clusters **5-1** and **5-2** consist of two molecules of germabenzenylgermylene **6** and a single additional Ge atom (Fig. 2b). Because it is known that a dibromodigermene ([R(Br)Ge]$_2$) is in equilibrium with a bromogermylene [R(Br)Ge:] in solution[28–30], the initial formation of **6** is possible although it is likely a transient species and was not detected by [1]H NMR spectroscopy during the course of the reaction. The structures of **5-1** and **5-2** are similar to each other except for the connecting positions between the Ge2 atom and the GeC$_5$ ring. The transformation of **5-1** to **5-2** can be explained by a [1,3]-sigmatropic rearrangement (on C3-C4-C5-Ge2 moiety, see Fig. 2b), consistent with the observation that exposure to ambient light is required to induce isomerization.

To gain further insight into this unexpected reactivity, the chemistry of germyl anion **4** was investigated (Fig. 3, eq. i). Compound **4** itself is thermally stable based on heating experiments in toluene-$d_8$ solution. On the other hand, the addition of [Tbb(Br)Ge]$_2$ (**2-Ge**) to isolated **4** in $C_6D_6$ at room temperature afforded **3-Ge**. However, it should be noted that the conversion of **4** was incomplete at this stage. Analogous to the reaction of **1** with **2-Ge**, heating was necessary for the reaction of **4** with **2-Ge** to achieve full conversion to **3-Ge** and Ge/C cluster **5-1**. As noted above, exposure to ambient light for 3 h gave

**Fig. 2 | Reaction of potassium germabenzenide 1 with 1,2-dibromodigermene** 2-Ge. **a** Products and the yields. Yields marked with [#] were determined by [1]H NMR spectroscopy. **b** Structures of Ge/C cluster **5-1** and **5-2**. Thermal ellipsoids were plotted at 50% probability. Hydrogen atoms and Tbb groups were omitted for clarity.

**Fig. 3 | Reactivity of germyl anion** 4 **and germabenzenylgermylene NHC complex** 6·NHC. NMR yields are marked with [#]. Reaction of **4** with **2-Ge** gave germabenzene **3-Ge** and cluster **5-1** which isomerized to **5-2** by ambient light (eq. **i**) while the reaction in the presence of NHC afforded **6·NHC** (eq. **ii**). Thermolysis of **6·NHC** forms corresponding germabenzene **3-Ge** (eq. **iii**).

**5-2.** When the reaction was conducted in the presence of 1,3-diisopropyl-4,5-dimethylimidazol-2-ylidene (Im$^{iPr2Me2}$), germabenzenylgermylene NHC adduct **6·NHC** was formed (Fig. 3, eq. ii). This result strongly suggests the formation of an intermediary germabenzenylgermylene **6** in this chemistry.

The independent preparation of **6·NHC** was accomplished by the reaction of the NHC complex of [Tbb(Br)Ge:] with **1** to give a yellow-orange solid in 87% yield (Fig. 3, eq. iii). Single crystals were obtained from a benzene/hexane solution, and the molecular structure of **6·NHC** was determined by X-ray crystallographic analysis. Heating a C$_6$D$_6$ solution of **6·NHC** at 110 °C for 12 h resulted in the complete consumption of **6·NHC** and the formation of germabenzene **3-Ge** (Fig. 3, eq. iii). The chemical shifts assigned to the Im$^{iPr2Me2}$ unit were also changed from those of **6·NHC** but were different from those of the free carbene Im$^{iPr2Me2}$, suggesting the formation of an NHC complex of zero-valent germanium atom(s). However, the structure of NHC-related compound has not yet been determined. Overall, all experimental results strongly suggest that **3-Ge** is not formed from **4** but from **6**.

## Mechanistic investigation of the formation of germabenzene 3-Ge

The potential energy surface of the model compound of **6**, Gebzl(H) Ge: (**6a**, Gebzl = 1-germabenzenyl), was explored using GRRM[31] calculations to shed light on the formation mechanism of germabenzene **3-Ge**. It gave 341 local minima and 303 transition states including only one chemically and energetically allowed isomerization pathway of **6a** via **TS1a-TS4a** (maximum $\Delta E$: + 16.5 kcal mol$^{-1}$ relative to **6a**, see Supplementary Fig. 39). It was determined that the Ge atom of the germabenzenyl ring was exchanged for that of the germylene in four steps to form the germabenzene-bridged germylene **INT4a**. It should be noted that the single step pathway from **6a** to **INT4a** via **TS5a** with retaining the original germabenzenyl ring (hydrogen transfer to germabenzene moiety) was also located in GRRM calculations, but this path is unfavorable due to its high energy barrier (+41.4 kcal mol$^{-1}$ relative to **6a**). Similarly, the barrier for the aryl transfer reaction in the phenyl-substituted model, Gebzl(Ph)Ge: (**6b**), was calculated to be +37.1 kcal mol$^{-1}$ (Supplementary Fig. 40), and thus Tbb transfer can be ruled out in this system.

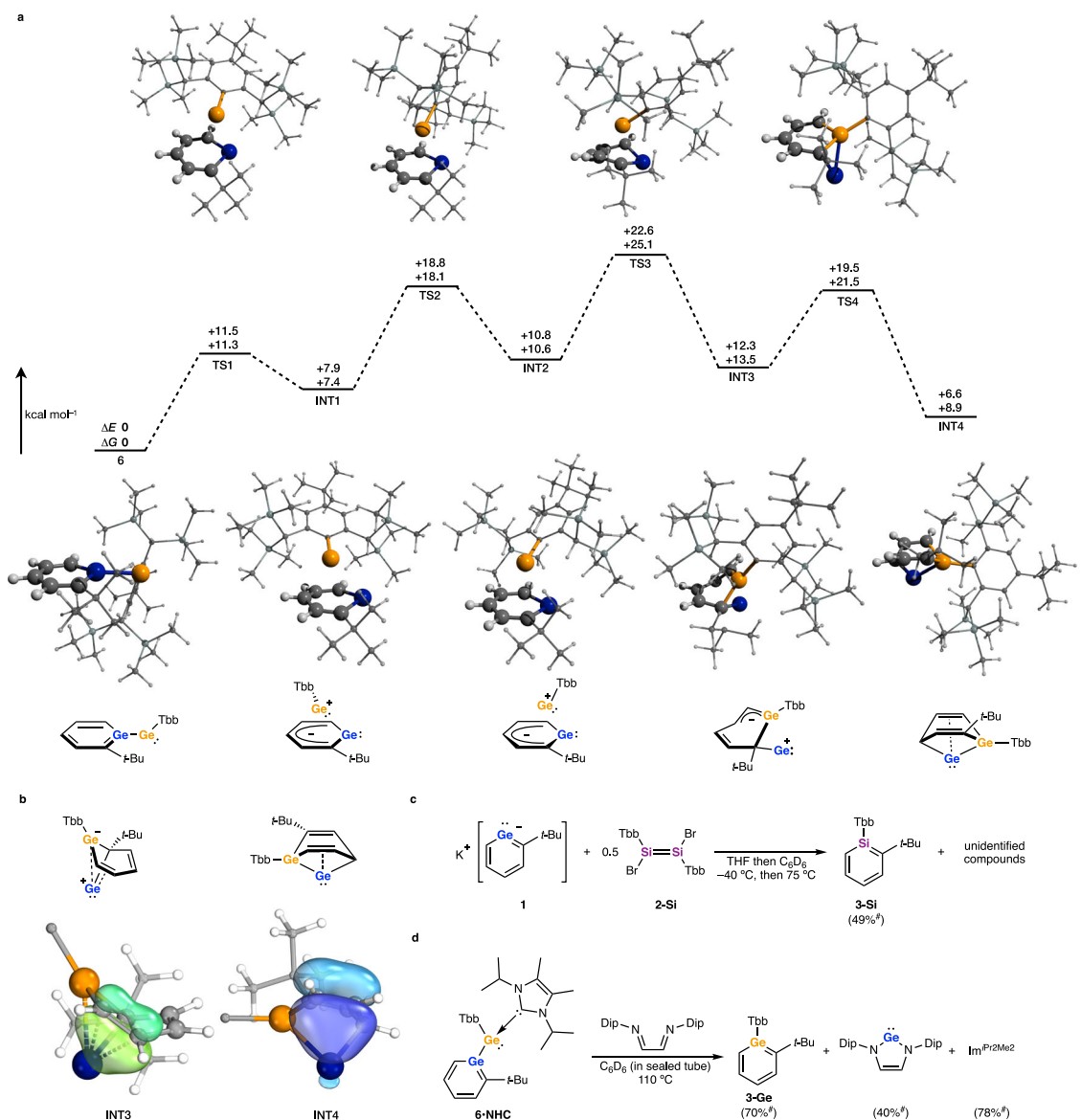

**Fig. 4 | Validation of isomerization pathway of germabenzenylgermylene** 6. NMR yields are marked with #. **a** Energy diagrams of the isomerization pathway of **6** at the B3LYP-D3/6-31 + G(2df,p) level of theory. **b** Intrinsic bonding orbitals of **INT3** and **INT4**. Electron delocalization among C−C−Ge is indicated. **c** Reaction of potassium germabenzenide **1** with dibromodisilene **2-Si** affording silabenzene **3-Si**. **d** Thermolysis of **6·NHC** in the presence of diimine.

The isomerization path optimization of non-truncated molecule **6** based on the results of GRRM calculations was also successfully found (Fig. 4a). The maximum energy barrier computed ($\Delta G = 25.1$ kcal mol⁻¹ relative to **6**) appears to be too high for isomerization to progress at room temperature. Experimentally, the formation of **3-Ge** is in competition with the formation of **4**, and the isomerization path of **6** is not necessarily favorable. However, concomitant KBr elimination during the formation of **6** and aromatization by Ge atom extrusion from **INT4** are considered to be exothermic. We presume that these steps supply the driving force necessary for this isomerization pathway. It should also be considered that Ge atom transfer from **INT4** is reasonable because it can restore aromaticity through Ge atom extrusion. The Ge/C cluster **5** appears to be formed from this Ge atom with two molecules of germabenzenylgermylene **6**.

It is notable that strong homoconjugative interactions between the outer Ge atom and a C = C bond are present in **INT3** and **INT4**. Natural bonding orbital (NBO) analysis revealed large second order perturbation energies of the $\pi_{C=C} \rightarrow$ Ge interactions in **INT3**

(87.6 kcal mol⁻¹) and **INT4** (59.2 kcal mol⁻¹), respectively. Furthermore, delocalization of two electrons among the Ge−C−C moiety were observed within the HOMO−1 of **INT3** and HOMO−4 of **INT4** (Supplementary Figs. 54 and 55) as well as in the intrinsic bonding orbitals (IBO)[32,33] of each compound (Fig. 4b). These types of interactions inducing homoaromaticity are widely accepted in general organic chemistry[34] and have also been observed in silyl- or germyl cations[35, 36] and metallylenes[37–39].

To confirm that the Ge atom on the germabenzenyl ring is indeed exchanged, the reaction of **1** with the 1,2-dibromodisilene **2-Si** was investigated (Fig. 4c). While it was not possible to isolate the intermediates of this reaction due to a lack of crystallinity of the products, the formation of silabenzene **3-Si** was confirmed spectroscopically in 49% yield under the same experimental conditions as the reaction of **1** with **2-Ge**. Importantly, no germabenzene **3-Ge** was observed. The formation of silabenzene **3-Si** was confirmed on the observation of a characteristic pattern corresponding to the silabenzene ring in the ¹H NMR spectrum as well as the characteristic chemical shift in the ²⁹Si NMR spectrum (81.1 ppm)[40].

This reaction clearly demonstrates a Ge atom replacement on the germabenzenyl ring with a Si atom, a net Ge atom transfer reaction. Although it is difficult to experimentally confirm the exchange in the aforementioned reaction with a digermene, a similar exchange of germanium atoms between germabenzenyl and germylene moieties is likely as proposed by the DFT calculations. Related heteroatom exchange reactions between aromatic compounds have been observed previously. For example, the synthesis of phosphabenzene from pyrylium by using P(SiMe₃)₃ or P(CH₂OH)₃ is proposed to proceed in this way[41,42]. However, there is almost no example of equivalent reactivity for group 14 elements. Müller and co-workers reported the Ge to Si exchange reaction of a germoldiide giving a silole[39], but this reaction proceeds with dearomatization. To the best of our knowledge, the reactions reported here are the first examples of an aromatic-to-aromatic nuclear exchange reaction on a benzene ring consisting only of group 14 elements.

The capacity of **6·NHC** to act as a source of Ge atoms was further demonstrated by the thermolysis of **6·NHC** in the presence of diimine [(DipN=CH−)₂, Dip = 2,6-diisopropylphenyl] (Fig. 4d). Heating a C₆D₆ solution in a sealed tube at 110 °C for 5 days gave the corresponding *N*-heterocyclic germylene (NHGe)[43] and the free carbene, Im^iPr2Me2. The research groups of Wesemann[17] and Iwamoto[18] have reported the silicon atom transfer reactions of **A** or **C** to diimines, resulting in the formation of an *N*-heterocyclic silylene (Fig. 1a). The formation of NHGe clearly indicates a Ge atom transfer from **6·NHC** to the diimine, consistent with the mechanism proposed herein.

To summarize these results, a possible reaction mechanism can be proposed in which the first step is the formation of germabenzenylgermylene **6** from potassium germabenzenide **1** and bromogermylene [Tbb(Br)Ge:]. While the additional equivalent of **1** to **6** gives germyl anion **4**, the reverse pathway from **4** to **6** also is possible. Compound **6** can isomerize to **INT4** with the driving force of strong homoconjugative interaction between the outer Ge atom and the C=C bond via four steps. Finally, **INT4** releases the Ge atom to form germabenzene **3-Ge**, and the extruded Ge atom is captured by two molecules of germabenzenylgermylene **6** to give **5-1** (Fig. 5).

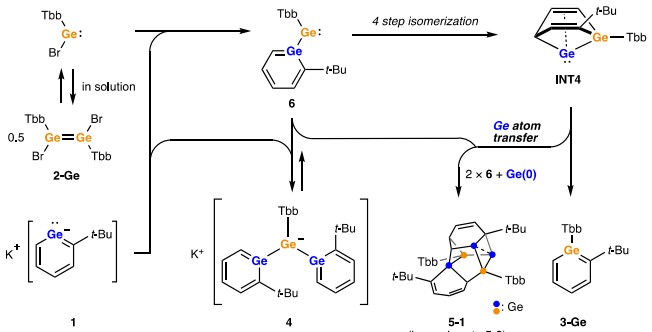

**Fig. 5 | Formation mechanism of germabenzene 3-Ge and cluster 5-1 in the reaction of potassium germabenzenide 1 with dibromodigermene 2-Ge.**

## Germanium transfer to bromogermylene: formation of [1.1.1] propellane skeleton

We also carried out the reaction of potassium germabenzenide **1** with 1,2-dibromodigermene **2-Ge** in hexane rather than THF, which resulted in the formation of another germanium cluster compound, 2,4,5-tribromopentagerma[1.1.1]propellane **7** (Fig. 6a). The propellane skeleton has long been studied due, in part to interest in the interactions between the bridgehead positions and relevant [1.1.1]propellane skeletons composed solely of Ge atoms have been reported in ref. 44 and ref. 45. To a hexane solution of 1.25 eq. (2.50 eq. as bromogermylene) of **2-Ge**, **1** was added at room temperature. Due to the insolubility of **1** in hexane, the complete consumption of **1** was observed visually after a few minutes. The ¹H NMR spectrum of the reaction mixture indicated the formation of **3-Ge** along with multiple unidentified compounds that are considered to be reaction products of Ge atom(s) and bromogermylene(s) [Tbb(Br)Ge:]. The reaction mixture converged to **3-Ge** and **7** after heating to 60 °C for 12 h. The structure of **7** was determined by the X-ray crystallographic analysis (Fig. 6b). The formation of **7** can be explained by the slow consumption of **1** due to the poor solubility in hexane. The slow formation of germabenzenylgermylene **6** causes the capture of the Ge atom of **INT4** by [Tbb(Br)Ge:], rather than by **6** as observed in THF.

The formation of **7** clearly indicates that the Ge atom is not always transferred to **6**. In addition, the products containing naked Ge atoms such as **5** and **7** are attracting increased interest due to their similarity with the elemental germanium. These experimental results suggest the potential availability of this reactivity as a unique synthetic method for the controllable formation of molecular germanium clusters bearing naked Ge atoms, the germanium analog of siliconoids[46].

We have demonstrated a germanium atom transfer reaction involving potassium germabenzenide **1** and 1,2-dibromodigermene **2-Ge**. This methodology has already been applied to yield two germanium clusters, **5** and **7**, which contain naked Ge atoms. The first step of the reaction is the formation of germabenzenylgermylene **6**, which isomerizes into germabenzenyl ring bridged germylene **INT4**. It is proposed that Ge atom extrusion occurs from **INT4** to also give germabenzene **3-Ge**. Compound **6** was isolable as an NHC derivative **6·NHC** and indeed demonstrated to act as a Ge(0) source in the reaction with a diimine. It was also found that the Ge atom of the germabenzenyl ring of **1** was replaced by the Ge or Si atom of **2**. This unprecedented aromatic-to-aromatic nuclear exchange reaction is expected to be utilized as a novel synthetic method for heavy benzene derivatives and may also provide access to new, unsaturated molecular germanium clusters as well as heavy benzenes.

## Methods

### Reaction of potassium 2-*tert*-butylgermabenzenide 1 with 0.5 eq of [Tbb(Br)Ge]₂ (2-Ge) in THF

To a THF (3 mL) solution of [Tbb(Br)Ge]₂ (**2-Ge**, 30.1 mg, 0.0250 mmol) was added a THF (3 mL) solution of **1** (11.6 mg, 0.0498 mmol) at −40 °C and stirred for 1 h at room temperature. After all solvents were removed in vacuo, the resultant brown solid was dissolved in C₆D₆ and then transferred to J. Young NMR tube. To this solution, 5.0 μL of 1,4-dioxane was added as an internal standard. ¹H NMR spectrum indicated the formation of germabenzene **3-Ge**

**Fig. 6 | Reaction of 1 with 1.25 eq. of 2-Ge in hexane. a** Products and the yields. NMR yields are marked with #. **b** Thermal ellipsoid plots of **7** at the 50% probability level. Tbb groups were omitted for clarity.

(17%), germyl anion **4** (36%), Ge/C cluster **5-1** (27%), and unidentified compounds. The yields were calculated as the conversion ratio based on **1**. Heating the mixture at 75 °C for 12 h increased the yields of **3-Ge** (50%) and **5-1** (37%) with the full consumption of **4**. Subsequent exposure to the ambient light for 5 h gave the mixture of **3-Ge** (50%) and **5-2** (37%). After filtration and removal of solvents in vacuo, crystallization from hexane afforded **5-2** (7.3 mg, 0.0048 mmol, 19% based on **1**) as blue-green crystals; m.p. 207–209 °C; Anal. Calcd. for $C_{66}H_{124}Ge_5Si_8$: C, 52.65; H, 8.30. Found: C, 52.42, H, 8.44; HRMS (DART): Calcd. for $C_{66}H_{125}Ge_5Si_8$ $[M+H]^+$: 1505.4055; found: 1505.3957 $[M+H]^+$; $^1H$ NMR (600 MHz, $C_6D_6$, 348 K): $\delta$ 0.22–0.36 (s + s + s + s, 72H), 1.15 (s, 9H), 1.21 (s, 9H), 1.31 (s, 9H), 1.34 (s, 9H), 2.15 (br s, 2H), 2.67 (br s, 2H), 3.07 (dd, 1H, $J$ = 8.4, 1.2 Hz), 3.10 (dd, 1H, $J$ = 3.6, 3.6 Hz), 4.97 (d, 1H, $J$ = 9.0 Hz), 5.69 (ddd, 1H, $J$ = 10.2, 7.2, 3.6 Hz), 5.96 (dd, 1H, $J$ = 10.2, 3.6 Hz), 6.20 (ddd, 1H, $J$ = 9.0, 8.4, 1.2 Hz), 6.49 (d, 1H, $J$ = 7.2 Hz), 6.87 (s, 2H), 6.88 (t, 1H, $J$ = 8.4 Hz), 7.00 (s, 2H); $^{13}C$ NMR (150 MHz, $C_6D_6$, 348 K): $\delta$ 1.8 (q), 1.9 (q), 2.2 (q), 3.0 (q), 30–32 (br), 31.27 (q), 31.30 (q), 33–34 (br), 33.1 (q), 34.3 (s), 34.5 (s), 36.9 (q), 37.5 (s), 38.2 (d), 40.7 (d), 44.6 (s), 62.2 (d), 122.8 (two signals are overlapped, d + d), 126.7 (d), 128.0 (d), 128.3 (d), 130.2 (d), 132.6 (d), 136.0 (s), 138.4 (s), 149.4 (s), 149.9 (two signals are overlapped, s + s), 151.4 (s), 159.5 (s), 183.6 (s). All signals were assigned by 2D-NMR (COSY, HSQC, HMBC, NOESY) techniques. The signals at 128.0 and 128.3 are undetectable in the $^{13}C\{^1H\}$ NMR spectrum due to overlapping with the signals of $C_6D_6$. These signals were observed and assigned by $^{13}C$ DEPT 135 and various hetero nuclear 2D-NMR techniques.

### Reaction of potassium 2-*tert*-butylgermabenzenide 1 with 0.5 eq of [Tbb(Br)Si]₂ (2-Si) in THF

To a THF (3 mL) solution of [Tbb(Br)Si]₂ (**2-Si**, 20.0 mg, 0.0179 mmol) was added a THF (3 mL) solution of **1** (8.3 mg, 0.036 mmol) at –40 °C, and the combined solution was stirred for 1 h at room temperature. After all solvents were removed in vacuo, the resultant brown solid was dissolved in $C_6D_6$, then transferred to J. Young NMR tube with the addition of 5.0 μL of 1,4-dioxane as an internal standard. The $^1H$ NMR spectrum showed the formation of silabenzene **3-Si** and unidentified compounds. Heating the mixture at 75 °C for 12 h resulted in the color changing to deep purple and the formation of 49% silabenzene **3-Si**. Attempts to isolate **3-Si** by recrystallization and sublimation were unsuccessful, and the structure of **3-Si** was confirmed by the following NMR signals (Supplementary Figs. 30–33). $^1H$ NMR (400 MHz, $C_6D_6$, 298 K): $\delta$ 0.10 (s, 18H), 0.17 (s, 18H), 1.30 (s, 9H), 1.46 (s, 9H), 2.52 (s, 2H), 6.77 (dd, 1H, $J$ = 9.2, 7.6 Hz), 6.95 (s, 2H), 7.23 (d, 1H, $J$ = 12.0 Hz), 7.87 (dd, 1H, $J$ = 12.0 Hz, 7.6 Hz), 7.96 (d, 1H, $J$ = 9.2 Hz); $^{29}Si$ NMR (79 MHz, $C_6D_6$, 298 K): $\delta$ 81.1.

## Data availability
All data generated or analyzed during this study are included in this Article and its Supplementary Information files. The X-ray crystallographic coordinates for structures reported in this study have been deposited at the Cambridge Crystallographic Data Centre (CCDC), under deposition numbers 2246032 (**3**), 2246033 (**4·18c6**), 2268057 (**5-1**), 2246034 (**5-2**), 2246035 (**6·NHC**), and 2246036 (**7**). These data can be obtained free of charge from The Cambridge Crystallographic Data Centre via www.ccdc.cam.ac.uk/data_request/cif. The coordinates of the optimized structures are present in the source file. Source data are provided with this paper.

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

## Acknowledgements

This work was supported by JSPS KAKENHI Grant Numbers JP19H05635 (N.T. and Y.M.), JP20K20447 (N.T.), JP19H05528 (N.T.), JP18H01963 (Y.M.), and JP16H04110 (N.T.) and Integrated Research Consortium on Chemical Science (IRCCS). Y.M. gratefully acknowledges ISHIZUE 2022 of Kyoto University. This study was supported by the Joint Usage/Research Center [JURC, Institute for Chemical Research (ICR), Kyoto University] by providing access to a Bruker Avance III 600 NMR spectrometer. We are furthermore grateful for the computation time, which was provided by the Super Computer Laboratory (ICR, Kyoto University). Elemental analyses were carried out at the Microanalytical Laboratory of the ICR (Kyoto University). The authors thank Prof. Masahiro Yamanaka (Rikkyo Univ.) for the helpful discussion about computational studies. Preliminary X-ray diffraction data of **5-2** and **6·NHC** were collected at the BL02B1 beamline of SPring-8 (JASRI, 2022A1621 and 2022A1200).

## Author contributions

Y.M., R.N., and N.T. determined the research strategy, and R.N. and R.S. performed the synthetic experiments. R.N. collected the physical properties and spectral data of all compounds appearing in this paper. R.N. and Y.M. performed the X-ray crystallographic analyses and theoretical calculations. Y.M., N.T., and R.W. supervised the work. All authors co-wrote the paper.

## Competing interests

The authors declare no competing interests.
