## [Peer Review File · Nature Communications]

Unusual Nuclear Exchange within a Germanium-containing Aromatic Ring that Results in Germanium Atom TransferReviewers' Comments:

Reviewer #1:

Remarks to the Author:

I very much enjoyed the chemistry described in "Unusual Nuclear Exchange within a Germanium-containing Aromatic Ring that Results in Germanium Atom Transfer" by Tokitoh, Mizuhata and co-workers.

What the authors report is a germanium atom transfer reaction employing Tokitoh's marvellous germabenzoyl anion and 1,2-dibromodigermene. This reaction leads to a germabenzene with concomitant germanium atom transfer. The almost fantastic aspect of this reaction is that the transferred Ge atom is the one from the original germabenzoyl anion, whereas the Ge atom in the formed germabenzene originates from the digermene. Consequently, analogous reaction employing the respective 1,2-dibromodisilene leads to the formation of a silabenzene. The fate of the released germanium atom is dependent on reaction conditions. In THF reaction with two equivalents of a germabenzoyl germylene gives cluster compound 5. Running the reaction in hexane leads to Ge cluster with [1.1.1]propellane skeleton.

The characterization of the compounds (NMR and structural analysis) is very good and also computational analysis of the Ge atom transfer process is carried out and explain very well.

As for the mechanistic pathways of cluster formation, the authors are less forthcoming. While I think the propellane formation can be understood as the linking of three Tbb-bromogermylene units with two Ge atoms, it would be nice to hear about the authors ideas about the formation of cluster 5.

What I find really odd, is that the authors do not reference Breher's pentagermapropellane: (doi: 10.1002/ange.200805289). This is not really acceptable. There is also no reference to Sekiguchi's spirobis(pentagerma[1.1.1]propellane, which is another compound structurally similar to 7.

Scheshkewitz siliconoids are mentioned at least.

Minor points:

- Figure 1 features compounds A, B C, and D. Only to compounds A and C reference is made on page 7.
- In the titles of Refs 1 and 2 is should read: Si-Pb and C-Sn
- In several Refs Muller should be changed to Müller

Overall, this is a very nice paper, featuring a couple of exciting compounds After fixing the points raised above, I am inclined to support publication.

Reviewer #2:

Remarks to the Author:

Norihiro Tokitoh, Yoshiyuki Mizuhata and co-workers present an interesting manuscript on low valent germanium chemistry. Starting point of the investigations is the chemistry of the germabenzoyl-anion 1. The anion was reacted with TBB substituted dibromodigermene 2. After heating a mixture of 1 and 2 two compounds 3 and 5 was observed.

At low temperature reaction between 1 and 2 in thf affords formation of 4. In the experimental part the yield of 4 is 71%, in Fig. 2 the yield is of 4 is 36%. Furthermore, the ratios for the formation of 4 are different. To this referee this is confusing and should be changed.

Reaction of 4 with further equivalents of 2 yields 3 and 5. In the experimental part the yield is different than in Figure 2b. Please change.

What is the product of the low temp reaction between 4 and 2, why did the authors change the temp. to 75°C and expose the mixture for three hours to light. What is the product without light?

Please explain the exposure to light.

Formation of 6NHC from 2 and 1 and NHC: Fig2b, 91%, Experimental part 87%. Please change.

The authors discuss a germanium atom transfer. In their discussions the germanium atom comes from

the germabenzoyl anion. This was further demonstrated with the formation of 3Si. However, why is a germanium atom coming from the digermene 2 excluded. A tbb transfer at the reaction temperatures necessary for the formation of 5 and 7 should be discussed. Please explain the exclusion of an aryl group transfer.

The authors present interesting chemistry, and the manuscript could be acceptable. To this referee in Fig 1b the transient species 6 is confusing. In Fig 2 6 plays no role and was never detected. Please try to rewrite the manuscript and present the nice chemistry in a more understandable way.

\

Thank you for inviting us to submit a revised version of our paper. Please find here our point-by-point responses to the questions/comments provided by the referees. A new version has been submitted with track changes.

Response to reviewer 1:

1. As for the mechanistic pathways of cluster formation, the authors are less forthcoming. While I think the propellane formation can be understood as the linking of three Tbb-bromogermylene units with two Ge atoms, it would be nice to hear about the authors ideas about the formation of cluster 5.

Response: We appreciate your comment. We have tried to uncover the mechanism of formation for cluster 5 from both experimental and computational viewpoints. Although the following new findings on the mechanism were obtained, the whole mechanism is extremely complicated, and no clear conclusions could be drawn.

We determined the structure of an intermediate for cluster 5 (the original compound number of 5 was changed to 5-2, and the newly characterized intermediate was assigned as 5-1). Compound 5-1 is an isomer of 5-2 and consists of two germabenzylgermylene 6 and a single Ge atom. We suspect that 5-1 was formed by capturing a Ge atom by two molecules of 6, while the formation of Ge₅-propellane 7 can be rationalized by the reaction of three bromogermynes with two Ge atoms. The isomerization mechanism from 5-1 to 5-2 can be explained by a [1,3]-sigmatropic rearrangement. This isomerization mechanism can explain the requirement for exposure to ambient light to yield 5-2. We have added this discussion to page 3.

2. What I find really odd, is that the authors do not reference Breher's pentagermapropellane: (doi: 10.1002/ange.200805289). This is not really acceptable. There is also no reference to Sekiguchi's spirobis(pentagerma[1.1.1]propellane, which is another compound structurally similar to 7. Scheschkewitz siliconoids are mentioned at least.

Response: We apologize for our incomplete citations and lack of confirmation. We did in fact have the citations in your comment, but they were excluded due to misuse of the citation tool at the last stage of editing. We added these two references as refs. 44 and 45 and the related sentences below on page 8.

(added to page 8, revised)

The propellane skeleton has long been studied due, in part to interest in the interactions between the bridgehead positions and relevant [1.1.1]propellane skeletons composed solely of Ge atoms have been reported by Breher⁴⁴ and Sekiguchi.⁴⁵

3. Figure 1 features compounds A, B C, and D. Only to compounds A and C reference is made on page 7.

Response: We appreciate your comment. We did make the reference to compounds B and D, but the order was inappropriate. We reorganized the reference numbers as below for clarity (refs. 16 and 17 were changed):

A (Germanium): ref. 15

Krebs, K M., Hanselmann, D., Schubert, H., Wurst, K., Scheele, M. & Wesemann, L. Phosphine-stabilized digermavinylidene. *J. Am. Chem. Soc.* **141**, 3424-3429 (2019).

A (Silicon): ref. 17 → **ref. 16**

Wilhelm, C., Raiser, D., Schubert, H., Sindlinger, C P. & Wesemann, L. Phosphine-stabilized germasilylenylidene: Source for a silicon-atom transfer. *Inorg. Chem.* **60**, 9268-9272 (2021).

B: ref. 16 → **ref. 17**

Wang, Y., Tope, C A., Xie, Y., Wei, P., Urbauer, J L., Schaefer, H F. & Robinson, G H. Carbene-stabilized disilicon as a silicon-transfer agent: Synthesis of a dianionic silicon tris(dithiolene) complex. *Angew. Chem. Int. Ed.* **59**, 8864-8867 (2020).

C: ref. 18

Koike, T., Nukazawa, T. & Iwamoto, T. Conformationally switchable silylone: Electron redistribution accompanied by ligand reorientation around a monatomic silicon. *J. Am. Chem. Soc.* **143**, 14332-14341 (2021).

D: ref. 19

Chen, M., Zhang, Z., Qiao, Z., Zhao, L. & Mo, Z. An isolable bis(germylene)-stabilized plumbylone. *Angew. Chem. Int. Ed.* **62**, e202215146 (2023).

4. In the titles of Refs 1 and 2 is should read: Si-Pb and C-Sn

Response: We appreciate you bringing this to our attention. We have corrected refs. 1 and 2 as indicated.

5. In several Refs Muller should be changed to Müller

Response: We appreciate you bringing this to our attention. We have corrected refs. 36-39 as indicated.

Response to reviewer 2:

1. At low temperature reaction between 1 and 2 in thf affords formation of 4. In the experimental part the yield of 4 is 71%, in Fig. 2 the yield is of 4 is 36%. Furthermore, the ratios for the formation of 4 are different. To this referee this is confusing and should be changed.

Response: We apologize for the confusing descriptions. Compound 4 could be synthesized independently by the reaction of 1 with 0.25 eq. of 2-Ge. The yield of 71% in Supporting Information is an isolated yield of 4 in this reaction, while the yield of 36% is an NMR conversion yield in the reaction of 1 with 0.5 eq. of 2-Ge (in Fig. 1a). The NMR yields of each step were added to the experimental part of the manuscript. The corresponding heading in Supporting Information was changed as follows for clarity.

(original, page S3)

Preparation of potassium bis(2-(*tert*-butyl)germabenzenyl)germylide 4.

↓

(revised, page S3)

Independent preparation of potassium bis[2-(*tert*-butyl)germabenzenyl]germylide 4 by the reaction of potassium germabenzenide 1 with 0.25 eq. of 1,2-dibromodigermene 2-Ge.

2. Reaction of 4 with further equivalents of 2 yields 3 and 5. In the experimental part the yield is different than in Figure 2b. Please change.

Response: As you pointed out, the yields in Supporting Information were incorrect. We have corrected them as follows.

(original, page S4)

Heating the mixture at 75 °C overnight and subsequent exposure to room light for 3 h gave the mixture of 3-Ge and 5 by the following yields based on the ¹H NMR; 3-Ge: 42%; 5: 53%.

↓

(revised, page S4)

Heating the mixture at 75 °C overnight and subsequent exposure to ambient light for 3 h gave the mixture of 3-Ge and 5 by the following yields based on the ¹H NMR; 3-Ge: 41%; 5: 57%.

3. What is the product of the low temp reaction between 4 and 2, why did the authors change the temp. to 75°C and expose the mixture for three hours to light. What is the product without light? Please explain the exposure to light.

Response: We appreciate for your comment. We characterized a structure of the intermediate for Ge/C cluster **5**, which persists in the absence of light, by X-ray analysis. Because it is an isomer of **5**, we changed the compound number of **5** to **5-2**, and the newly characterized isomer was assigned as **5-1**. The structural difference between these two compounds is bonding situation between the GeC₅ ring and Ge₂ atom (Please see Fig. 2b on the revised manuscript). The isomerization mechanism is explained by a [1,3]-sigmatropic rearrangement. We believe the exposure to ambient light induces this step. We have added this discussion to page 3 and revised Fig. 2.

4. Formation of 6NHC from 2 and 1 and NHC: Fig2b, 91%, Experimental part 87%. Please change.

Response: We appreciate your comment. The yield of 87% is correct. We changed the Fig. 2 (Fig. 3 on the revised manuscript).

5. The authors discuss a germanium atom transfer. In their discussions the germanium atom comes from the germabenzene anion. This was further demonstrated with the formation of 3Si. However, why is a germanium atom coming from the digermene 2 excluded. A tbb transfer at the reaction temperatures necessary for the formation of 5 and 7 should be discussed. Please explain the exclusion of an aryl group transfer.

Response: We appreciate your comment. As you pointed out, there was a lack of explanation regarding the exclusion of aryl transfer reactions. The discussion in the original manuscript was based on the fact that direct hydrogen transfer is unfavorable in the hydrogen-substituted model. We have now found that in the phenyl-substituted model compound, the aryl transfer reaction retaining the original germabenzene skeleton has almost the same high barrier as in the hydrogen-substituted system (37.1 kcal mol⁻¹). Although we could not find a suitable pathway in the Tbb-substituted system, the aryl transfer process from **6** can be ruled out based on these results.

We have rewritten the description of the calculation part (pages 5-6) in detail and added a sentence below and Fig. S38 for the results of calculations using the phenyl-substituted model.

(added to page 5, revised)

Similarly, the barrier for the aryl transfer reaction in the phenyl-substituted model, Gebzl(Ph)Ge: (**6b**), was calculated to be +37.1 kcal mol⁻¹ (Fig. S38), and thus Tbb transfer can be ruled out in this system.

6. The authors present interesting chemistry, and the manuscript could be acceptable. To this referee in Fig 1b the transient species **6** is confusing. In Fig 2 **6** plays no role and was never detected.

Response: We appreciate you bringing this to our attention. Actually, **6** was not detected experimentally. We changed Fig. 1.

7. In SI, Geph -> Gebzl

Response: We appreciate you bringing this to our attention. We have made the appropriate correction.

Other major changes:

Minor corrections have been made throughout the manuscript and supporting information. Major changes are listed below.

- SI, page S4: The information for the isolation of Ge/C cluster **5-1** was added.
- SI, pages S7-8: Original Table S1 was split into two tables (S1 and S2), and the data for compound **5-1** was added to Table S1.
- SI, page S12: As for the results of the crystallographic structural analysis of **5-1**, Fig. S3 and Table S5 were added.
- SI, pages S18-19: The NMR spectra of **5-1**, Figs. S10-S14, were added.
- SI, pages S27-28: Figs. S26 and S27 were added to show the details of the reaction.

Reviewers' Comments:

Reviewer #1:

Remarks to the Author:

I checked on the revised version of the "Unusual Nuclear Exchange within a Germanium-containing Aromatic Ring that Results in Germanium Atom Transfer" manuscript. It is nice to see that attempts to solve the puzzle of the formation of cluster 5 (now 5-2) have been undertaken.

As the authors have properly reacted to all the raised issues from my side, and have also made other improvements, I have no reservation to recommend publication now.

Reviewer #2:

Remarks to the Author:

The authors have addressed my concerns and to this referee the manuscript is ready for publication.

Minor point: please exchange reference 17 on the first page line 34 by 16.

Thank you for inviting us to submit a final revision of our paper. Please find here our point-by-point responses to the questions/comments provided by the referees. A new version has been submitted with track changes.

Response to Reviewer #2:

1. Minor point: please exchange reference 17 on the first page line 34 by 16.

Response: Thank you for pointing this out. We have corrected it.